# Discovering Research Hypotheses in Social Science using Knowledge Graph Embeddings

Rosaline de Haan[1], Ilaria Tiddi[2], and Wouter Beek[1]

[1] Triply, Amsterdam, The Netherlands
{rosaline.de.haan,wouter}@triply.cc
[2] Vrije Universiteit Amsterdam, Amsterdam, The Netherlands
i.tiddi@vu.nl

**Abstract.** In an era of ever-increasing scientific publications available, scientists struggle to keep pace with the literature, interpret research results and identify new research hypotheses to falsify. This is particularly in fields such as the social sciences, where automated support for scientific discovery is still widely unavailable and unimplemented. In this work, we introduce an automated system that supports social scientists in identifying new research hypotheses. With the idea that knowledge graphs help modeling domain-specific information, and that machine learning can be used to identify the most relevant facts therein, we frame the problem of hypothesis discovery as a link prediction task, where the ComplEx model is used to predict new relationships between entities of a knowledge graph representing scientific papers and their experimental details. The final output consists in fully formulated hypotheses including the newly discovered triples (hypothesis statement), along with supporting statements from the knowledge graph (hypothesis evidence and hypothesis history). A quantitative and qualitative evaluation is carried using experts in the field. Encouraging results show that a simple combination of machine learning and knowledge graph methods can serve as a basis for automated scientific discovery.

**Keywords:** Scientific Discovery · Knowledge Graphs · Link Prediction · Social Science

## 1 Introduction

Scientific research usually starts with asking a question, followed by doing background research, and then formulating a testable hypothesis. Doing background research to properly substantiate a hypothesis can be a difficult and time-consuming task for scientists. It is estimated that over 3 million scientific articles are published annually, a number that keeps growing of 4% each year [25]. The fast rate at which new publications appear, as well as the inefficient way in which scientific information is communicated (e.g. PDF documents), calls for more efficient data analysis and synthesis, in a way that scientists formulating new research hypotheses can be supported rather than overloaded.

The task of significantly speeding up the steps in the scientific process is generally called automated scientific discovery [15]. The latest years have seen Artificial Intelligence approaches for automated scientific discovery in various

scientific fields, either relying on symbolic knowledge representation or machine-driven methods. Knowledge graphs such as the Gene Ontology[1] and the ontology collection of the Open Biological and Biomedical Ontology Foundry[2] have been used to encode domain-specific information, such as representing biological systems from the molecular to the organism level. Machine Learning and particularly link prediction methods, that help predicting which missing edges in a graph are most likely to exist, have also been used, e.g. to support medical scientists by showing them new associations between drugs and diseases [15, 19].

There is currently not much automated support for social scientists when it comes to getting new insights from scientific information. This is partly due to the more qualitative and uncertain nature of social science data, making it hard to represent, and consequently less machine-interpretable [3]. One effort in this direction is the COoperation DAtabank (CODA), where an international team of social scientists published a structured, open-access repository of research on human cooperation using social dilemmas. The dataset represents about 3,000 research publications with their experimental settings, variables of observation, and quantitative results. Given the large amount of structured information available, and the success of predictive methods seen in other disciplines, it is natural to think that a hybrid method could be designed, to automatically suggest social scientists new hypotheses to be tested.

Here, we study the problem of automatic hypothesis discovery in the field of social sciences. We propose to frame the problem as a link prediction task, and particularly to exploit the structured representation of the domain to learn research hypotheses in the form of unseen triples over a knowledge graph describing research papers and their experimental settings. Using the knowledge graph embeddings learnt with the ComplEx model, we predict the likelihood of new possible relationships between entities, consisting in the variables studied social science research. These relationships are then used to provide the experts with new research hypotheses structured in a *statement* (the newly predicted associations), *evidence* and *history* (both triples existing in the graph). We quantitatively and qualitatively assess this approach using experts in the field, which help us evaluating the accuracy and meaningfulness of the discovered hypotheses. Our main contributions can be summarised as follows: (i) we show how a thorough structured representation of scientific knowledge can help supporting the automatic discovery of research hypotheses; (ii) we present a preliminary approach combining knowledge graph data and machine learning to help experts in formulating new research hypotheses; (iii) we show how our method can be applied in the field of social science.

## 2   Related Work

Our work relates to three areas, namely (i) existing methods for representing and mining scientific knowledge, (ii) approaches for automated hypothesis discovery in science and (iii) knowledge graph embedding methods and applications.

---

[1] http://geneontology.org/

[2] http://www.obofoundry.org/

*Representing and Mining Scientific Knowledge.* Several methods have been developed to represent scientific knowledge and foster interoperability and reproducibility. Micro- and nanopublications [4, 8] have been introduced in the last decade as standardised formats for the publication of minimal scientific claims, i.e. minipublications. Such models allow to describe evidence and nuanced scientific assertions expressing a relationship between two predicates (e.g. a gene relates to a disease), together with provenance information describing both the methods used to derive the assertion and publication metadata. The DISK hypothesis ontology [7] was introduced to capture the evolution of research hypotheses in the neuroscience field, including the provenance and revisions. More precisely, a DISK hypothesis consists of structured assertions (hypothesis statement), some numerical confidence level (hypothesis qualifier), the information of the analysis that were carried out (hypothesis evidence), and prior hypotheses revised to generate the current one (hypothesis history). In the field of medical science, the different elements to be included in a hypothesis can be described with the PICO ontology[3], describing Patients, the Condition or disease of interest and its alternative (Intervention), and the Outcome of the study.

Repositories for storing scientific publications at large scale in the form of knowledge graphs include both domain-specific initiatives (e.g. the AI-KG [5] for Computer Science and the Cooperation Databank [22] for the social sciences), and domain-independent projects such as the Open Research Knowledge Graph (ORKG) project[4]. These initiatives focus on representing research outputs in terms of their content, i.e. describing approach, evaluation methods, results etc., rather than publication context such as year, authors and publication venues. This type of novel representations allows to automatise not only the search for new research, but also to compare it at large scale.

Some work has focused on developing systems that aid with mining claims in the existing literature. The AKminer (Academic Knowledge Miner) system [9] was introduced to automatically mine useful concepts and relationships from scientific literature and visually present them in the form of a knowledge graph. Similarly, [17] uses text-mining to automatically extract claims and contributions from scientific literature and enrich them through entity linking methods. Supervised distant learning was used by [14, 24] to extract PICO sentences from clinical trial reports and support evidence-based medicine.

*Machine-supported Hypothesis Discovery.* Automated hypothesis discovery using intelligent systems has been interest of study for a long time. Earliest work include the ARROWSMITH discovery support system [21] to help scientists in finding complementary literature for their studies and formulate a testable hypothesis based on the two sets, and the work of [1], which used various machine learning techniques to discover patterns, co-occurrences and correlations in biological data. These approaches inspired the work of [20], which relies on a scientific text collection to discover hypotheses, via Medical Subject Headings (MeSH)-term based text-mining.

---

[3] https://linkeddata.cochrane.org/pico-ontology
[4] https://www.orkg.org/orkg/

Biomedical literature was also used by [10] to develop a link discovery method based on classification, where concepts are learnt and used as a basis for hypothesis generation. An Inductive Matrix Completion method was presented by [12], where the discovered gene-disease associations where supported by different types of evidence learnt as latent factors. The Knowledge Integration Toolkit (KnIT) [11] used methods such matrix factorization and graph diffusion to reason over a network of scientific publications in the biomedical field to generate new and testable hypotheses. The work of [15] shows how scientific insights can be generated using machine support also in the field of astronomy and geosciences. Their model allows to create multiple variants of hypothesised phenomena and their corresponding physical properties; these are matched in the existing empirical data, and scientists can both refine them and use them to justify a stated research hypothesis. The DISK ontology was also used in the field of neuroscience for automated hypothesis assessment [6].

*Knowledge Graph Embeddings for Link Prediction.* Machine learning methods for knowledge graph completion (or link prediction) use inductive techniques, mostly based on knowledge graph embeddings or rule/axiom mining, to locally and logically predict the likelihood of certain link between two nodes to exist [13]. Currently, the tensor decomposition ComplEx method [23] has proven to be the most stable in terms of performance and scalability [2]. Link prediction methods have been previously used for hypothesis discovery. Authors of [16] first create a knowledge graph from biomedical data and then convert it to a lower dimensional space using graph embeddings. The learnt embeddings are then used to train a recurrent neural network model to predict new drug therapies against diseases. A similar approach is the one of [19] to generate hypotheses on re-purposing drugs for rare diseases; the method relies on graph embeddings learnt over a large knowledge graph including information from the literature of pharmacology, genetics and pathology.

## 3   Background and Motivating Scenario

*The COoperation DAtabank Knowledge Graph.* The COoperation DAtabank consists in ∼3,000 studies from the social and behavioural sciences published in 3 languages and annotated with more than 300 cooperation-related features, including characteristics of the sample participating in the study (e.g. sample size, average age of sample, percentage of males, country of participants), characteristics of the experimental paradigm (structure of the social dilemma, incentives, repeated trial data), and quantitative results of the experiment (e.g. mean levels of cooperation, variance in cooperation, and effect sizes). The dataset was designed to be fully compliant with the F.A.I.R. principles, and has been published as an openly available knowledge graph[5] to allow domain experts to perform their analyses in minutes, instead of many months of painstaking work [22].

Before continuing with the knowledge graph structure, we need to familiarise the reader with the basic concepts of experimental science. Studies using this

---

[5] http://data.cooperationdatabank.org/

methodology may observe a relation between two (one independent, one dependent) variables, which can be quantified as an effect size (representing the quantitative result). The goal of the single experiments carried within in a study is to test whether the dependent variable (DV) changes for when modifying the value of the Independent Variable (IV), which indicates there is a relationship between the two variables. In the case of the Databank, one could imagine an experiment aimed at studying the impact (effect size) of a person's social values (independent variable) over her willingness to cooperate (dependent variable). With this in mind, the CODA knowledge graph includes publications consisting of a `cdo:Paper` class that links to an arbitrary set of `cdo:Study`, i.e. experiments performed in different settings and with different goals. Additional metadata about the paper such as publication date, authors etc. are included as properties of a `cdo:DOI` class. Each `cdo:Study` links to one or more conditions tested, represented by the class `cdo:Observation`, that are in turn modelled as comparisons of one or two different `cdo:Treatment`.

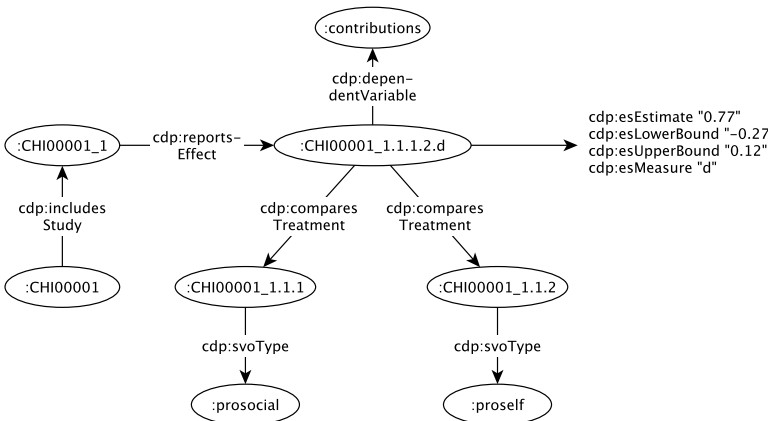

**Fig. 1.** Example of an observation comparing prosocial vs. proself behaviour.

In a practical example, Figure 1 shows the paper `:CHI00001` including the study `:CHI00001_1`, which in turns reports the observation `:CHI00001_1.1.1.2.d` comparing treatment `:CHI00001_1.1.1` and `:CHI00001_1.1.2` (we call them T1 and T2 for simplicity). Treatments consist in the experimental settings that the experimenter modifies with the goal of testing how and if the cooperation between participants of a game varies significantly. In our example, the experimenter manipulated the property `cdp:svoType` which, recalling what stated above, consists then in the independent variable observed. This is confirmed by the fact that T1 and T2 have a different value for the property (`:prosocial` and `:proself` respectively). Similar to `cdp:svoType`, any RDF property whose domain is the class `cdo:Treatment` is organised in a domain-specific taxonomy of independent variables, representing information relative to cooperation in social dilemmas. Finally, in order to represent how and how much the cooperation varies during an observation/experiment, we use the class `cdo:DependentVariable` for the DV

and the datatype property `cdp:esEstimate` for the effect size measurement, e.g. `CHI00001_1.1.1.2.d` measures the DV `:contributions` and its effect size has a value of 0.77[6]. The positive effect size reported by the experimental observation means that T1 scored higher on cooperation than T2, indicating that participants with a pro-social value orientation showed a more cooperative behaviour than participants who had a pro-self value orientation.

*Challenge and Proposed Solution.* In the scenario above, it is natural to see how the CODA knowledge graph intrinsically represents research hypotheses that were tested in the human cooperation literature. In other words, one can consider each `cdo:Observation` subgraph as a research hypothesis that aims at testing whether there exists a relation between the `cdo:IndependentVariable` and `cdo:DependentVariable`. The effect size value of each observation then tells us the strength of such relation, identified by the experiment performed to validate the hypothesis. The question we ask is therefore whether it is possible to learn new, plausible observations starting for the representations recorded in CODA and, more in general, how to support domain experts in producing new research hypotheses through a more automated method. While such methods have been widely presented in the biomedical field, applications facilitating automated support for the social scientists have yet to be implemented. The solution we propose is to frame the problem of learning research hypotheses as a link prediction task, where we exploit the existing `cdo:Observation` subgraph structures to learn new unseen triples involving a `cdo:IndependentVariable` and `cdo:DependentVariable`. Our assumption is that entities and relationships neighbouring the predicted links could help completing the new research hypotheses. Following similar approaches in the biomedical field, we train a knowledge graph embedding model to predict the likelihood of a new possible association between an IV and a DV, and develop a system that suggests new possible research hypotheses including both triples existing in CODA and new predicted triples according to a predefined structure. We then quantitatively and qualitatively assess the accuracy and meaningfulness of the discovered hypotheses in a user-study based on the domain expertise of social scientists from the field.

## 4   Approach

The proposed approach includes three steps: a pre-processing phase for data selection and generation of the model input (Section 4.1), a learning phase including parameter tuning, model training, and link prediction (Section 4.2), and a last phase for the automated generation of hypotheses (Section 4.3).

### 4.1   Pre-processing

The first step is to choose the right amount of CODA information to retrieve, and create an input for the embedding model to be able to predict new triples.

---

[6] CODA contains two types of effect size measures, i.e. the correlation coefficient $\rho$ and the standardized mean difference $d$, which can be easily converted to one another. For simplicity, we will only refer to Cohen's $d$ values from now on.

*Observation Selection.* First, we define a set of criteria to select the CODA observations, namely:

1. instances of the class `cdo:Observation`;
2. observations reporting using Cohen's *d* as effect size measure;
3. observations comparing two treatments;
4. observations linking to an instance of a `cdo:DependentVariable`.

The SPARQL query used to get the observations can be found online[7], and results in 4,721 observations, the study, paper and DOI that reported them, the effect size with confidence levels, the experimental design, and sample size and standard deviation per treatment pair.

A further refinement is performed by analysing the independent variables of each observation. We identify the properties-values for which the two treatments compared by an observation differ on, e.g. `cdp:svoType/:prosocial` vs. `cdp:svoType/:proself` in the example of the previous section. To prevent noise and reduce complexity, we dropped observations that had no differing predicates (errors attributed to the large sparsity of the data and to human annotation), or that might differ for more than one property. This left 2,444 observations to train the model, coming from 632 papers and 858 studies, and including 128 unique IVs and 2 unique DVs.

*Data Permutation.* Since KG embedding methods are generally not capable of learning continuous variables, we learn effect sizes as categorical instead of continuous information. This is also motivated by the fact that Cohen's *d* is in fact a measure that can be interpreted categorically [18]. To this end, we created a new RDF property `cdp:esType` and a set of 5 instances of the class `cdo:ESType` that a `cdo:Observation` might point to, representing the 5 bins mapping the continuous effect size values to Cohen's categories[8]. Table 1 shows the ranges for each bin/instance, and their respective effect size types.

**Table 1.** Effect size ranges, their interpretation and the respective instance created.

| Effect size range | Intepretation | Instance |
|---|---|---|
| -infinity, -0.5 | large/medium negative correlation | `:largeMediumNegativeES` |
| -0.5, -0.2 | small negative correlation | `:smallNegativeES` |
| -0.2, 0.2 | no correlation | `:nullFinding` |
| 0.2, 0.5 | small positive correlation | `:smallPositiveES` |
| 0.5, infinity | large/medium positive correlation | `:largeMediumPositiveES` |

As also explained in Section 3, an effect size is an indication of the size of the correlation between an independent and a dependent variable, measured based on the different IV values that two treatments take during an experimentation.

---

[7] `https://data.cooperationdatabank.org/coda/-/queries/`
`link-prediction-selection-query`

[8] Due to the relatively small sets, medium and large effects were grouped together.

This means that, in order to predict a new correlation between IV and DV, one would have to predict multiple triples, i.e. at least one per treatment (and their respective IV values). In order to simplify the task, we summarise the factor that influences the effect size into a single node, by considering IV values pairs as single hypotheses. We therefore combine all possible values for a given IV property into pairs, assigning a hypothesis number to each pair, and create a new node that is linked to the original T1/T2 values through the property `cdp:hypothesis`. The new nodes, shown e.g. in Table 2, are then used for the hypothesis generation. For continuous properties reporting many different values in the object position, four different ranges were automatically created to prevent the generation of an excessive amount of hypotheses. Similarly, pairs with the same IV values in a different order were considered as the same hypothesis (e.g. T1=proself/T2=prosocial and T1=prosocial/T2=proself were both linked to `:SVOtypeH2`), but the effect size node of the observation was switched (positive to negative, or vice versa) to maintain the direction of the correlation coherent.

**Table 2.** Hypothesis nodes based on combinations of IV values for T1 and T2.

| IV | T1 value | T2 value | Hypothesis Node |
|---|---|---|---|
| SVO type | individualist | prosocial | `:SVOtype_H1` |
| SVO type | prosocial | proself | `:SVOtype_H2` |
| SVO type | individualist | altruist | `:SVOtype_H3` |

We then link the created hypothesis nodes to the dependent variable nodes using three new predicates, related to the type of correlation that is observed: `cdp:hasPositiveEffectOn`, `cdp:hasNoEffectOn`, `cdp:hasNegativeEffectOn`. These properties are based on the statistical significance of the observation, computed using the 95% confidence interval for the effect size. A confidence interval measures the imprecision of the computed effect size in an experiment. When the interval does not include 0, it can be inferred that the association is statistically significant ($p < 0.05$). In other words, the confidence interval tells us how trustworthy is the observation we are analysing, in terms of effect size, population estimate, and direction of the effect. Depending on the confidence interval, we use `:hasNoEffectOn` if the effect size is not significant, while `:hasNegativeEffectOn` and `:hasPositiveEffectOn` are used with observations indicating a significant negative and positive correlation between IV and DV, respectively. When no confidence interval was given in the data, we derive the direction of the correlation using the rule of thumb as reported of [18]: observations with an effect size below $-0.2$ got a negative effect property, observations that reported effect sizes above 0.2 got a positive effect property, and observations with an effect size between -0.2 and 0.2 got a no effect property. This led us to a total of 751 positive effect triples, 1,017 no effect triples and 676 negative effect triples.

*Dataset Creation.* The last step of the pre-processing task consists in the conversion into learnable subgraphs, i.e. sets of triples. To do this, we use part of the information already in the data, namely observation ID, the independent and dependent variables, the IV values for the two treatments, and combine them

with the computed effect size type, the hypothesis number, and relationship to the dependent variable. A construct query[9] was used to generate subgraphs as depicted in Figure 2 for 2,444 observations. This led to a dataset of 29,339 triples, that served as input for the link prediction model.

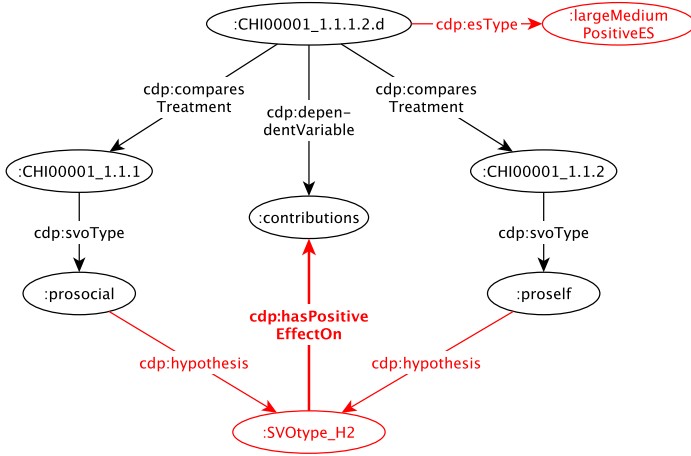

**Fig. 2.** New graph for the observation `CHI00001_1.1.1.2.d` used as input for the link prediction task. In red, the nodes and edges created. In bold, the link to be predicted.

### 4.2   Learning and Predicting Triples

*Training and Testing.* We use the created dataset to learn a model predicting unseen triples to be used in new hypotheses. Strictly speaking, the prediction consists in identifying triples including a hypothesis number, an effect size predicate and a dependent variable, e.g. ⟨`:SVOtype_H2 cdp:hasPositiveEffectOn :contributions`⟩. To do this, all triples reporting a negative or a positive effect were gathered. We decided not to make predictions for the no-effect triples, as experts might be less interested in non-interesting relations between variables to frame their hypotheses. Investigating this for future work could be interesting. From the total 1,427 effect triples, the 243 unique hypotheses in subject position, the 2 unique predicates and the 2 unique dependent variables in object position were used to learn how to generate new combinations. This yielded to $243 * 2 * 2 = 972$ total triples, of which 412 were already in the dataset and marked as "seen", while the other 560 were denoted as "unseen".

We then used the ComplEx model to learn the likelihood of each triple. We first split the dataset into a training set of 24,539 triples, a test set of 2,400 triples, and a validation set of 2,400 triples. A corruption strategy is then used to generate negative statements. Parameter tuning was finally performed to explore impact on the model performance, cfr. Table 3 for the final configuration.

---

[9] `https://data.cooperationdatabank.org/coda/-/queries/`
`Rosaline-Construct-Link-Prediction`

Standard metrics such as mean reciprocal rank, hits@N and mean rank were used to evaluate the trained model.

**Table 3.** Final parameter configuration.

| Parameter | Value |
|---|---|
| batches_count | 555 |
| epochs | 100 |
| $k$ (dimensionality) | 200 |
| $eta$ (# neg. samples generated per each pos.) | 15 |
| loss | multiclass_nll |
| embedding_model_params | {'negative_corruption_entities': 'all'} |
| LP regulariser params | {'p':1, 'lambda':1e-5} |
| Xavier initialiser params | {'uniform': False} |
| Adam optimizer params | {'lr': 0.0005} |

*Link Prediction.* The learnt model was used to compute ranks and scores for unseen triples. Ranks indicate the position at which the test set triple was found when performing link prediction, while scores are the returned raw scores generated by the model. Probabilities of unseen triples are also calculated by calibrating the model. We set a positive base rate of 0.5 (50%) to indicate the ratio between positive vs. negative triples. After calibration, a probability for each unseen triple was predicted. We then obtained their ranks, score and probabilities for the 560 unseen triples, to be later used during the hypotheses generation step. A sample of these is in Table 4 below.

**Table 4.** Prediction example of unseen triples.

| Statement | Rank | Score | Prob. |
|---|---|---|---|
| :iteratedStrategy_H6 cdp:hasPositiveEffectOn :cooperation | 1 | 7.38 | 0.98 |
| :iteratedStrategy_H9 cdp:hasPositiveEffectOn :cooperation | 2 | 7.32 | 0.98 |
| ... | ... | ... | ... |
| :uncertaintyTarget_H1 cdp:hasPositiveEffectOn :cooperation | 3816 | 0.10 | 0.19 |
| :exitOption_H1 cdp:hasNegativeEffectOn :contributions | 4659 | −0.03 | 0.17 |

### 4.3   Hypotheses Generation

The final step is to automatically generate human-interpretable hypotheses, based on the unseen triples predicted by the model. Each statement from Table 4 was converted into a readable text using a prefixed structure following the DISK ontology. A *hypothesis statement* was created by disassembling the triples into respectively the independent variable (the predicted subject), the type of effect (the predicted predicate) and dependent variable (the predicted object). The *hypothesis evidence* was created by querying the CODA knowledge graph for labels of both IVs and DVs, and by converting the effect type property into decapitalised words with spacing. We also retrieve the description of both the IV class and the relevant IV values. The *hypothesis history* was built by retrieving the DOIs of papers that studied that combination of IV values. An example of a generated hypothesis is shown below.

**Hypothesis Statement**

Partner's group membership has negative effect on contributions

**Hypothesis Evidence**

Dependent Variable (DV): `https://data.cooperationdatabank.org/id/dependentvariable/contributions`

Independent Variable (IV): `https://data.cooperationdatabank.org/vocab/prop/targetMembership`
*Whether the participant is interacting with a partner identified as ingroup, outgroup, or stranger.*

The IV values to compare in the treatments (T1, T2) are :

| Treatment | IV value | Description |
|---|---|---|
| T1 | ingroup | Partner(s) is a member of the participant's group |
| T2 | ingroup_and _outgroup | When an experimental treatment explicitly provides information that a partner or group belongs to both an ingroup and an outgroup |

**Hypothesis History**
`http://dx.doi.org/10.1016/j.joep.2013.06.005`
`http://dx.doi.org/10.1177/0146167205282149`
`http://dx.doi.org/10.1016/j.ijintrel.2011.02.017`

*Implementation.* The current approach was implemented using Python 3.7.7. The ComplEx model was implemented using the Ampligraph[10] library. All the code and results can be found on GitHub[11]. The queries were made using the SPARQL API service of the CODA knowledge graph, hosted by TriplyDB[12].

## 5   Evaluation

We first quantitatively and qualitatively evaluate the model performance through known metrics and inspection of the independent variable embeddings. We then evaluate the generated hypotheses through domain experts.

### 5.1   Model Performance

We used three different types of metrics as indication of how well the model was capable of predicting the triples in the test set: mean reciprocal rank (MRR), hits@N, and mean rank (MR). Reciprocal rank measures the correctness of a ranked triple, and mean RR is defined as $MRR = \frac{1}{|Q|} \sum_{i=1}^{|Q|} \frac{1}{rank_i}$, where $Q$ is the number of triples and $rank_i$ the rank of the $i$th triple predicted by the model. Hits@N indicates how many triples are ranked in the top N positions when ranked against corruptions, i.e.: $Hits@N = \frac{1}{|Q|} \sum_{(s,p,o) \in Q} ind(rank(s,p,o) \leq N)$ where $Q$ is the triples in the test set, $(s,p,o)$ is a triple $\in Q$, and $ind(\cdot)$ is an

---

[10] `https://github.com/Accenture/AmpliGraph`

[11] `https://github.com/roosyay/CoDa_Hypotheses`

[12] `https://coda.triply.cc/`

indicator function returning 1 if the positive triple is in the top $N$ triples, 0 otherwise. We use three values for $N$, namely 1, 3 and 10. Finally, the MR score is the sum of the true ranks divided by the total amount of ranks, defined as $MR = \frac{1}{|Q|} \sum_{i=1}^{|Q|} rank_{(s,p,o)_i}$. Note that the MR score is not robust to outliers, and is therefore only taken into account together with the other metrics. An overview of the model performance can be found in Table 5. Overall, these scores indicate a reasonable model performance but some room for improvement. Our approach only includes one type of model and one dataset, hence a more extended assessment should be considered in the future.

**Table 5.** Model performance evaluation metrics.

| MRR | Hits@10 | Hits@3 | Hits@1 | MR |
|-----|---------|--------|--------|--------|
| 0.68 | 0.75 | 0.69 | 0.64 | 279.91 |

### 5.2   Qualitative Analysis

To get insight into how the model effectively learnt the data, we created a visualisation of the main independent variables (see Figure 3). To do this, the 400-dimensional embeddings of 128 unique independent variables were retrieved from the trained model and transformed into an array of (128, 400). We used a UMAP reduction to reduce the 400 dimensions to 2 only, allowing then to display the embeddings in a 2-dimensional space. In order to find the optimal number of clusters in this space, we used an elbow method measuring the Within-Cluster Sums of Squares (WCSS) without finding any significant distinction. We therefore used a silhouette analysis, revealing that 23 clusters was the best balance between the number of clusters and a relatively high silhouette score (silhouette score=0.49). Clusters were obtained using scikit-learn's KMeans (K=23) and the visualisation was obtained using the Matplotlib package.

As shown in Figure 3, most clusters are groups of variables that are `rdfs:sub-ClassOf` the same class. For example, in clusters 3, 11 and 15, all the variables related to respectively punishment, emotion and leadership are clustered together. Clusters such as 2 and 13 seem to have less cohesion, as no overarching topics can be found that group IVs together. This can be due by a larger variety of the studies that analysed these IV, and potentially a lack of more data. Both clusters also include variables related to reward, showing that studies with reward-related IV are more heterogeneous and were not grouped together.

### 5.3   Domain Expert Evaluation

In order to qualitatively evaluate the generated hypotheses, 5 domain experts from the CODA team were asked to fill out a user-study. A Google form was created where the experts, after receiving information about the background and the goal of the study, were shown the 10 most likely and 10 most unlikely hypotheses predicted by the model. The 20 hypotheses were shown in a random order using the structure presented in Section 4.3. The experts were asked to indicate which 10 hypotheses they considered likely, and which 10 unlikely. A

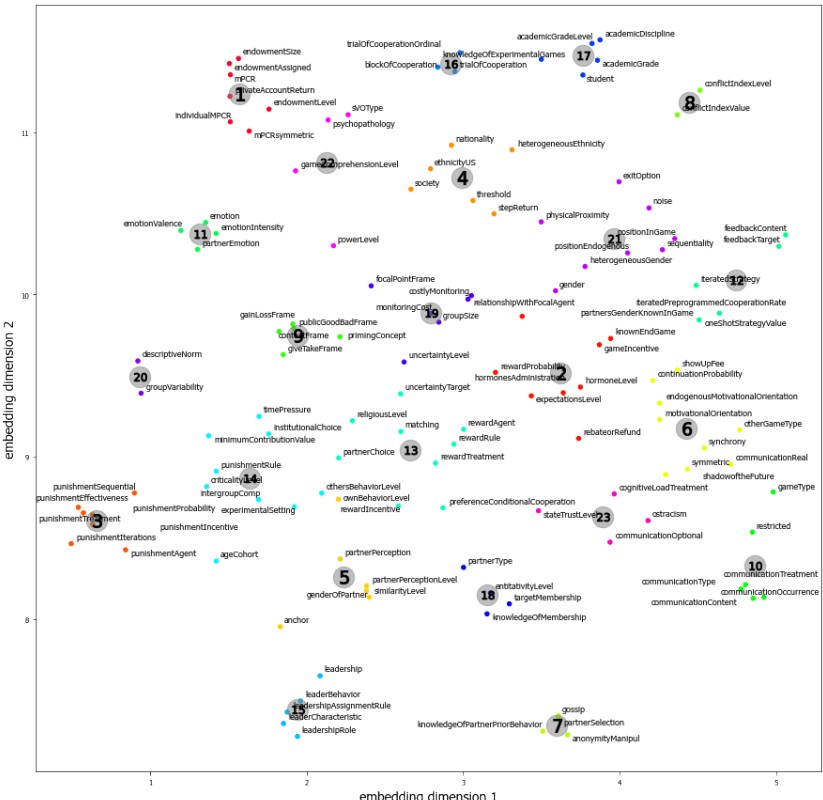

**Fig. 3.** Independent variables grouped in 23 clusters. Please cfr. Visualisation.ipynb on Github for better quality.

final part for remarks was also included.

Table 6 shows how the experts rated the likelihood of the hypotheses. These can be easily read as "[Hypothesis Statement] *when comparing* [T1 value] vs. [T2 value]", e.g. "SVO type has negative effect on cooperation when comparing a group of individualists vs. a group of competitors". Two hypotheses including miscellaneous IV values did not make sense according to the experts and were omitted. Overall, the majority of the experts rated 12 out of 18 hypotheses as the model did, while only 6 hypotheses were rated opposite of the model. Out of 5 experts, 2 rated more than 9 hypotheses the same as the model, which is higher than chance level, while the other 3 experts scored exactly on chance level. No experts scored below chance level. It should be noted that some experts took more time to fill the evaluation form, as they provided more details in the open-ended questions, and some variety could be seen in how experts rated the hypotheses. We relate this to the complexity of social science data, causing different perspectives to reach different conclusions. Looking at the overall average however, the experts rated 10 hypotheses the same as the model did. This shows that the model output is not random, and that similarities between the expert opinions and the model were found. More in general, we consider our results

encouraging enough to confirm the idea that a link prediction-based approach is a valuable method to predict hypotheses over structured data.

**Table 6.** Expert evaluation. #L and #UL refer to the number of experts scoring a hypothesis as likely and unlikely, respectively. *Pred.* indicates the model prediction.

| | Hypothesis Statement | T1 value | T2 value | #L | #UL | Pred. |
|---|---|---|---|---|---|---|
| 1 | MPCR has positive effect on contributions | $(-0.401, 0.3)$ | $(0.3, 0.5)$ | 3 | 2 | likely |
| 2 | partner's group membership has negative effect on contributions | ingroup | ingroup and outgroup | 0 | 5 | likely |
| 3 | intergroup competition has positive effect on contributions | individual group | intergroup competition | 4 | 1 | likely |
| 4 | anonymity manipulation has positive effect on cooperation | high | low | 3 | 2 | likely |
| 5 | time pressure has negative effect on contributions | time-pressure | time delay | 2 | 3 | likely |
| 6 | SVO type has negative effect on cooperation | individualist | competitor | 3 | 2 | likely |
| 7 | ethnicity (us) has positive effect on cooperation | white | black or african american | 1 | 4 | likely |
| 8 | iterated strategy has positive effect on cooperation | predominantly cooperative | other | 4 | 1 | likely |
| 9 | nationality has negative effect on contributions | JPN | AUS | 3 | 2 | unlikely |
| 10 | exit option has negative effect on contributions | 0 | 1 | 2 | 3 | unlikely |
| 11 | exit option has positive effect on contributions | 0 | 1 | 1 | 4 | unlikely |
| 12 | emotion has negative effect on cooperation | neutral | disappointment | 2 | 3 | unlikely |
| 13 | emotion has positive effect on cooperation | neutral | disappointment | 2 | 3 | unlikely |
| 14 | preference for conditional cooperation has negative effect on cooperation | freeriders | hump-shaped contributors | 4 | 1 | unlikely |
| 15 | uncertainty target has positive effect on cooperation | loss | threshold | 4 | 1 | unlikely |
| 16 | iterated strategy has positive effect on contributions | tit-for-tat | tit-for-tat+1 | 1 | 4 | unlikely |
| 17 | preference for conditional cooperation has positive effect on cooperation | freeriders | hump-shaped-_contributors | 1 | 4 | unlikely |
| 18 | uncertainty target has negative effect on cooperation | loss | threshold | 0 | 5 | unlikely |

## 6   Conclusions

We have introduced an approach to automatically support domain experts to identify new research hypotheses. The approach is based on a link prediction task over a knowledge graph in the social science domain, where new edges between nodes are predicted in order to create fully formulated hypotheses in the form of a hypothesis statement, a hypothesis evidence and a hypothesis history. The quantitative and qualitative evaluation carried using experts in the field has shown encouraging results, namely that a simple combination of machine learning and knowledge graphs methods can support designing more complex systems for the automated scientific discovery.

Improvements of our approach could be made as future work, namely by optimising the data modelling and the machine learning approach for social science data. Such type of data has in fact an uncertain nature, and missing information

can create an inner bias in the model and have implications for the results. Solutions to cope with such bias should be investigated. As mentioned, the approach should be also tested on datasets of different domains to see how it could perform. Some information from the data was lost due to binning continuous variables (including effect size values), and the learning task could be improved on this aspect. An end-to-end task could be envisioned to learn the subgraphs directly, instead of pre-processing them and reducing the task to predicting one link between two entities. Finally, our model generally predicted only links with the highest probabilities based on statistical frequency learnt from the data structure. An interesting avenue would be to investigate what makes an hypothesis interesting other than popularity, and how to learn them.

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
