# OpenReview forum: "Discovering Research Hypotheses in Social Science using Knowledge Graph Embeddings"
_eswc-conferences.org/ESWC/2021/Conference/Research_Track — ESWC 2021 Research_

### Official Review · AnonReviewer2 · 2021-01-08
**Review of Discovering Research Hypotheses in Social Science using Knowledge Graph Embeddings**

**Rating:** 1
**Confidence:** 4
**Impact:** 3
**Design And Technical Quality:** 4

**Review:**

This paper presents an approach to hypothesis generation modeled as a linked prediction task over the CODA knowledge graph in social sciences. The approach consists of pre-processing the graph in order to generate training, validation and  test splits of subgraphs which will then be used to train and evaluate a link prediction model. ComplEX is used as the underlying knowledge graph embedding algorithm to produce the encoding of the subgraph triples. No description of the link prediction model architecture is provided in diagram or textual form that helps understand how the link prediction task has been actually modeled, though. Table 3 is the closest we can see in this regard, but it is limited to an account of the hyperparameters used to train the model.  Looking at the code in the scripts available in GitHub (https://github.com/roosyay/CoDa_Hypotheses), I understand the authors directly used the predict method associated with the ComplEx model in the AmpliGraph library in order to obtain the score of the likeliness of the new triples according to the model previously fit against the KG using the generated splits.

The modeling choice to address this problem as a link prediction task is an interesting decision in this kind of setting where you start from some ground truth in the form of a structured knowledge graph. It is particularly useful that CODA is a well curated and apparently big enough resource for the task at hand. The dataset represents about 3,000 research publications with their experimental settings, variables of observation, and quantitative results. However, the authors constrain the analysis to the information that is already structured in the KG . It could be expected that the actual value is in the information that has not been curated yet, which lies in the scientific papers, precisely because it has not been curated yet and domain experts may need some assistance in doing so.

Related approaches, e.g. in medicine, do not focus on extracting triples from the KG as a source for their prediction. On the contrary, information is extracted from the actual, unstructured text of the scientific publications, producing triples (hypotheses) that are then ranked against the target KG using some model.  If the CODA KG is the beginning and the end of your analysis, how long do you think it will take until you reach a fixpoint and no new hypothesis can be proposed? Furthermore, if in the end what you are doing is one form of reasoning over the graph, why not having the domain experts model a number of inference rules to do that instead of resorting to a learning approach? In other words, how does hypothesis generation differ from other forms of inference using knowledge graphs for your problem? If you had a reasoner and a set of rules had been produced or modeled by an expert, would inference produce the same hypotheses you predict based on link extraction?

In the introduction, you mention generating hypotheses in social sciences is particularly harder compared to other fields. How would it be different implementation-wise, e.g. compared to medicine? Could you not have reused similar approaches used in those fields? And, if this is not possible (I guess not, given the formulation of the problem, which seems to depend on the actual way the CODA KG is represented, based on the connection between dependent and independent variables) how is your approach different? I think more abstraction would be needed here. For example, what do the correlation coefficient and mean difference mean for effect measuring?

In the related work section you mention nanopublications and DISK , but then the hypotheses you generate do not follow such representations. I wonder if this would have been desirable for the sake of interoperability. I miss mentions of approaches to represent scientific information and resources in machine-readable form like research objects, which are also related to nanopublications. In the area of machine-supported hypothesis discovery, there is a great deal of work, particularly applied to medicine, which should be mentioned. For example, the MOLIERE and, more recently, AGATHA systems, which have been extensively evaluated on large corpora like PubMedCentral and MEDLINE. As a matter of fact, I miss a table in the evaluation section comparing the results you obtain with your approach on your dataset with what could be done using algorithms like those proposed by those systems. On the other hand,  I see much of the cited related work has been produced by research in the semantic web community. IMO, this is a minimally viable coverage of the SotA in this area, which is broader, and the authors should also look at what is being done in other related communities.

Evaluation details are a little scarce. As mentioned, I would have liked to see a comparative with other approaches adapted to the CODA dataset. Also, a more fine-grained account of the results per predicate type (type of the effect: hasPositiveEffectOn or hasNegativeEffectOn) involved in the predicted triples would have helped to obtain a deeper understanding of the results reported in table 5. In addition to the experiments you do with social scientists on the quality of the research hypotheses you produce, do you have any metric of how "valuable" they turn to be in practice? Have you measured their potential to lead to new scientific discoveries? and if so, have you compared this against human performance? How dependent of the specific field is this? Would you be equally successful in another field? more? less?



**Anonymity:**

Yes, I would like my review to remain anonymous.

**Reuse And Availability:**

4: High

**Strong Points:**

- Interesting problem in an interesting domain.
- Thoroughly conducted implementation, which involves an important part of data pre-processing.
- Good application of existing resources (CODA) and KGE algorithms, although evidence is not provided that ComplEx is the best choice for this task.

**Subreviewer:**

I submitted this review.

**Weak Points:**

- Not sure of the value of extracting triples from the KG instead of the literature. In this case, why not modeling this problem as a classical rule-based inference problem?
- Evaluation is a little limited.
- Experimentation and comparison with other KGE algorithms would have been useful.

---

> ### Author Rebuttal · Authors · 2021-01-29
>
> Thank you for your extensive and enthusiastic review, which contains many interesting research questions and ideas. Many of these are large projects that can be considered for future work. In this paper, we focused on using a graph embedding method in a new and mostly unexplored field when it comes to hypothesis discovery.
>
> We decided to not explain ComplEx too much since the model isn’t novel, and we wanted to focus the attention on how to processing the CoDa data for hypothesis discovery. You also mention that CoDa is the beginning and the end of our analysis, meaning that we will reach a fixpoint and no new hypotheses that can be proposed. Because the dataset is relatively small, we were capable of generating all possible hypotheses (560 in total) and predicting their likelihood. This is something that could be solved by expanding the data using actual text of the scientific publications as you suggest. The advantage of using CoDa is actually that the content of papers is already represented in a structured/ machine readable format, allowing us to focus on the machine learning method rather than having to deal with the information retrieval step. We keep this as an idea for future work.
>
> Regarding comparison with classical rule based inference, comparing the symbolic and subsymbolic method (or try to combine them hybridly, too) is indeed an interesting idea. Notice that the CoDa graph had a rather complex structure, so the idea is that the subsymbolic representation of embeddings might allow to better predict new relationships when compared to a classical rule-based approach (which would also likely suffer in terms of computational costs).
>
> In your fourth paragraph you ask how the current problem differs implementation-wise from different fields (e.g. medicine). For this specific type of research the way in which the model input is curated matters a lot. Since you compare two groups or values from an independent variable and look at the difference in outcome between those groups or values on the dependent variable. You can’t simply link two variables together (as is often the case in the biomedical field). This makes our approach different in the sense that it caters specifically to this type of research.
>
> In the related work we describe a number of approaches that relate well to our specific topic. We tried to move away from the larger discovery systems for the biomedical domain. In the biomedical domain, the data structure and therefore the applied methods have developed differently then what is needed in the social science domain.
>
> Lastly, a more fine-grained result of different effect types was not possible due to the amount of data available in CoDa. Also we decided to not measure how valuable the hypotheses would be in terms of them leading to new scientific discoveries. The main reason for this being that we had a limited number of domain experts available to us, who only had limited time to help us evaluate the performance of the model, checking the validity of the model had priority in this regard. However it is still a valid question, as well as the additional questions you raised that could be very interesting to answer in the future.

---

> > ### Comment · AnonReviewer2 · 2021-02-03
> > **Thank you for your comments**
> >
> > Like I said, this is promising work. I am glad to see that you are aware of the limitations of the current approach and willing to make progress in such direction.

---

### Official Review · AnonReviewer1 · 2021-01-11
**Review of Discovering Research Hypotheses in Social Science using Knowledge Graph Embeddings**

**Rating:** 2
**Confidence:** 4
**Impact:** 4
**Design And Technical Quality:** 4

**Review:**

In this paper, the authors explore a methodology for building an automated system for supporting social scientists in identifying novel research hypotheses. This system relies on knowledge graph embeddings (trained with ComplEx) and through link prediction techniques they aim at connecting entities that could form a new hypothesis. These newly formed hypotheses (previously unconnected entities) will then be translated into statements accompanied by evidence and history (triples already existing in the graph) for the sake of explanation.

The approach is sound. From the creation of the knowledge graph to the evaluation.

However, I would like to flag some concerns and raise some questions.
In section 4.2, you state “We decided not to make predictions for the no-effect triples, as experts might be less interested in non-interesting relations between variables to frame their hypotheses. Investigating this for future work could be interesting.” I find it particularly fair to frame the research problem, however I would disagree with the interests of the experts for two reasons. I if were a social scientist, I would be interested in non-interesting, because: 1) if I trust 100% your system, I am aware that such route doesn't lead to any interesting result, and 2) if I don't trust your system, I might want to either prove or debunk such hypothesis. Actually, I think at this point, non-interesting is not the right label for this class of hypotheses.

This is more a typo. In section 5.1, when you explain the MRR, you write “where Q is the number of triples”. |Q| is the number of triples, Q is just the set of triples.

In section 5.2, you state “the 400-dimensional embeddings of 128 unique independent” but this seems to be inconsistent with table 3 reporting the configuration parameters. I am assuming here that the k (dimensionality) value reports the size of the embeddings which is 200. Is there a chance that I mistakenly made this connection?

In section 5.3, I am looking at the results in table 6. In such section you state “Overall, the majority of the experts rated 12 out of 18 hypotheses as the model did, while only 6 hypotheses were rated opposite of the model. Out of 5 experts, 2 rated more than 9 hypotheses the same as the model, which is higher than chance level, while the other 3 experts scored exactly on chance level.” Reading the table, I can confirm the first part: 12/18 correctly identified. However, the second part of the paragraph (Out of 5 experts …) is not evincible from the table. There is no info in the table that allows me to see that 2 experts rated more than 9 hypotheses the same as the model.

Another question on the same matter, it looks like the expert received 20 hypotheses (10 likely + 10 unlikely), but table 6 reports only 18 (8 likely + 10 unlikely)

This question instead is more on the adoption of ComplEx. Is there a reason why you choose this algorithm to train your embeddings? Knowing that there are many solutions available in the state of the art, what makes this algorithm the most suitable one? Are you planning to use other algorithms, just to put them in competition and see which one performs the better?

AFTER THE REBUTTAL:
I thank the authors for their reply and clarifying their stance.
Good work


**Anonymity:**

No, I would like my review to be deanonymized.

**Reuse And Availability:**

3: Medium

**Strong Points:**

The approach is sound. From the creation of the knowledge graph to the evaluation.

**Subreviewer:**

I submitted this review.

**Weak Points:**

The paper does not seem to exhibit clear weak point. If I have to highlight one, I can point out the fact that authors choose to stick with one training algorithm

---

> ### Author Rebuttal · Authors · 2021-01-29
>
> Thank you for your review on our paper, you address some interesting points. First, thank you for pointing out the typo. We will correct this. Secondly, we would like to explain why we chose to not address the no-effect triples. The goal of this paper was to come up with new hypotheses. This means that you look for meaningful links that might also predict something in practice, based on this we decided to leave those out. However, if the goal would have been different (e.g. giving social scientists insight into the field),  then we agree that it would also be interesting to include the no-effect triples as well.
>
> You mention that you can not read from the table that 2 of 5 experts rated more than 9 hypotheses the same as the model. This was also not intended to be in the table, as that would make the table even larger than it already is. Therefore we just described this finding in text.  Furthermore, in the paper it is mentioned that two hypotheses were omitted because these contained miscellaneous independent variable values.
>
> We chose the tensor decomposition ComplEx method because it has proven to be the most stable in terms of performance and scalability [2]. In the future we would like to explore different approaches as well and compare these.
>
> 2. Bianchi, F., Rossiello, G., Costabello, L., Palmonari, M., Minervini, P.: Knowledge Graph Embeddings and Explainable AI (4 2020). https://doi.org/10.3233/SSW200011

---

### Official Review · AnonReviewer4 · 2021-01-13
**Interesting application of link prediction in a novel domain, but preliminary evaluation**

**Rating:** 1
**Confidence:** 4
**Impact:** 3
**Design And Technical Quality:** 3

**Review:**

The paper proposes a methodology based on link prediction for identifying new research hypothesis in the field of sociology.

The paper reads well and presents an interesting line of research. The claim that the method is able to produce fully formulated hypotheses seems excessive considering that the approach mainly produces possible associations between variables and the description of those variables in a knowledge base.
The state of the art is quite comprehensive.

The application of link prediction technique of hypothesis generation is not a new idea, but the paper applies it for the first time to the sociology domain.
I am not completely convinced about the step for producing ‘hypothesis evidence’ and the resulting ‘evidences’ discussed in the paper. I do not understand how simply displaying labels of IVs and DVs could be characterized as evidence, especially if compared to the evidences that are typically produced by logical inference. For instance, in the example at page 11, the ‘evidences’ do not seem to specifically support the conclusion. Maybe ‘evidence’ is the wrong term? They seem just to be some auxiliary information. I do not see evidences that the model has good explanation capabilities.

The evaluation is the weaker part of the paper and may be too preliminary for a conference paper. In the quantitative evaluation the presented methods are not evaluated versus any other alternative. It would have been interesting to compare it to other state-of-the-art approaches for link detection based on graph embeddings (e.g., TransE, DistMult) or other hypothesis detection solutions adopted in other domains (e.g., Biomedical). The user study with domain experts is interesting, but there seems to be a quite strong disagreement between experts, especially considering that they are only evaluating the top 10 most likely and unlikely hypotheses according to the system. What is the average agreement? Are the presented approach assessments statistically comparable to the expert ones? Also, it is not clear why Table 6 reports only 18 of the mentioned 20 hypotheses.
It seems that in addition to the fact that an hypothesis is likely or not it would be important the assess also the novelty or importance of the hypotheses. Otherwise it is hard to claim that the method is able to produce “new” research hypotheses. On a positive note, I appreciate that all the data and code are shared via GitHub.

In conclusion, the paper presents an interesting analysis of the application of a link prediction approach in a new domain, but the weak evaluation makes it a borderline contribution.


**Anonymity:**

Yes, I would like my review to remain anonymous.

**Reuse And Availability:**

3: Medium

**Strong Points:**

- Good application of link prediction in a novel domain
- Well-written paper
- Good state of the art
- The evaluation data are available


**Subreviewer:**

I submitted this review.

**Weak Points:**

- Excessive claim in the introduction
- The method is not evaluated against any alternative
- The user study is a bit superficial

---

> ### Author Rebuttal · Authors · 2021-01-29
>
> Thank you for your thorough review. Your point about the hypotheses evidence is a valid one, especially from the perspective of a logician. We will think about naming this differently.
>
> You also mention some points about the evaluation. The goal of the evaluation was to see if the model was capable of distinguishing likely and unlikely hypotheses similarly to experts. Since we had a total of 560 hypotheses we decided to go with the most likely and unlikely ones. Furthermore, in the paper it is mentioned that two hypotheses were omitted because these contained miscellaneous independent variable values.
>
> We would also like to restate what we wrote for the review of anonReviewer2:
> “Also we decided to not measure how valuable the hypotheses would be in terms of them leading to new scientific discoveries. The main reason for this being that we had a limited number of domain experts available to us, who only had limited time to help us evaluate the performance of the model, checking the validity of the model had priority in this regard. However it is still a valid question, as well as the additional questions you raised that could be very interesting to answer in the future.”
>
> Lastly, in the context of our data, the hypotheses we discovered can be considered as new, as the data was about one specific topic (human cooperation) and curated with the goal of becoming a complete as possible data base for this topic. If further advancements are made in information representation in the social science field on a larger scale, the method proposed in our paper could be very effective in discovering new hypotheses for a broader scope as well.

---

### Official Review · AnonReviewer3 · 2021-01-15
**Research hypothesis discovery using KG embeddings**

**Rating:** 2
**Confidence:** 5
**Impact:** 4
**Design And Technical Quality:** 4

**Review:**

The authors frame the problem of hypothesis discovery  as a link prediction task, where the ComplEx model is used to predict new relationships between entities of a knowledge graph representing scientific papers and their experimental details. Specific contributions may be summarized as follows:

 (i) they how how a
thorough structured representation of scientific knowledge can help support
the automatic discovery of research hypotheses;
(ii) They present a preliminary approach combining knowledge graph data and machine learning to help experts
in formulating new research hypotheses;
(iii) they show how the method can be applied to social science meta-research.

The contributions are sufficiently original and clearly described. I recommend accept.

**Anonymity:**

Yes, I would like my review to remain anonymous.

**Reuse And Availability:**

4: High

**Strong Points:**

--The paper is well written and tackles an important problem
--Evaluation and validation seem adequate, but could be supplemented further in future work
--The work seems relevant to the semantic web


**Subreviewer:**

I submitted this review.

**Weak Points:**

I don't have major weaknesses to point out, although I think the paper can do with some light proofreading

Minor typos:

On page 5, the sentence 'The goal of the single experiments carried within in a study...' seems awkward and ungrammatical, pls rewrite.

---

> ### Author Rebuttal · Authors · 2021-01-28
>
> Thank you for your positive review. We will proof read the paper again and remove the grammatical issues.

---

### Official Review · AnonReviewer5 · 2021-01-17
**The authors applied ComplEx knowledge embedding on around 30K triples related to cooperation databank to do link prediction of research hypotheses for social science.**

**Confidence:** 3
**Impact:** 2
**Design And Technical Quality:** 2

**Review:**

Automatically generating research hypotheses is an important and timely topics given the ever-increasing number of published articles. But how to generate an effective hypothesis is really hard. In this research, authors gave a try by using the cooperation databank triples and knowledge embedding methods to predict potential research questions and did the expert evaluation.

But this paper has several issues:
- no novelty in algorithm design
- no novel perspective on applying algorithm to real world problem. Authors should explore related methods, rather than just pick CompIE. Authors might want to compare the related methods on knowledge embedding (https://github.com/thunlp/OpenKE) to pick up the most effective one. Simply choosing CompIE can be arbitrary. It is unclear whether the proposed method is effective and reusable, or rather just a practice or exercise.

**Anonymity:**

Yes, I would like my review to remain anonymous.

**Rating:**

-2: Reject

**Reuse And Availability:**

2: Low

**Strong Points:**

- great applications of applying knowledge embedding to generate research hypotheses, for researches related to clinical trials and meta-analysis, or medical domains, can be interesting and practical
- simple evaluation by the domain experts on the predicted hypotheses
- clearly stated the whole procedure, easy to follow

**Subreviewer:**

I submitted this review.

**Weak Points:**

- novelty in algorithm design: it is hard to know the novelty in algorithm design, as authors just applied one existing methods
- ComplEx was published in 2015, there are already several latest ones, such as RotatE (https://arxiv.org/abs/1902.10197).
- did not compare with other related methods (https://github.com/thunlp/OpenKE)
- evaluation result is not promising, half are unlikely

---

> ### Author Rebuttal · Authors · 2021-01-28
>
> Thank you for your review on our paper. We would like to address your most important topics.
>
> The focus of our paper was indeed not on designing a novel algorithm, and neither on comparing existing algorithms. We chose the tensor decomposition ComplEx method because it has proven to be the most stable in terms of performance and scalability [2]. The application of the ComplEx algorithm in the social science domain is novel and hard due to the specific nature of the data, which contains a lot of uncertainty.
>
> We would also like to clarify the last point, “evaluation result is not promising, half are unlikely”. This is inherent to the setup of our evaluation. As the output of our model is the likelihood of hypotheses, we evaluated the model by letting experts check whether they agreed with the eight most likely and the ten least likely results. The outcome of this evaluation was that the majority of experts rated 12 out of 18 as the model did, which is a promising result.
>
> 2. Bianchi, F., Rossiello, G., Costabello, L., Palmonari, M., Minervini, P.: Knowledge Graph Embeddings and Explainable AI (4 2020). https://doi.org/10.3233/SSW200011

---

### Decision · Program_Chairs · 2021-02-23

**Decision:**

Accept with shepherding

**Comment:**

There was broad consensus among the reviewers that this paper tackles an important problem with interesting ideas; the applications of the work are broadly useful; the paper is well written; and the contribution is relevant to the semantic web community. While there is some concern about the (algorithmic) novelty of the work, the authors have explained that the novelty lies in the application of the algorithm to social sciences data for the first time. I recommend to 'accept with shepherding' to give authors a chance to respond to reviewers and revise the paper accordingly.
Specific issues that have to be addressed are the following:
- clarify better the scope of the paper. Some reviewers' concerns are focused on the fact that the algorithm has been applied with one approach only. This is a clear limitation. However, it seems that the contribution of the paper is not the specific ML technique adopted in the method, but the method itself. This should be extremely clear.
- although this depends on the previous point, it is somehow independent: clarify what is the novelty introduced by the solution proposed by the authors. This can be done by (i) framing the problem by pointing out clear research questions and hypothesis; ((ii) clearly positioning the paper with respect to the state of the art.
- make the evaluation more robust by adding details that are currently missing, such as the experimentation and comparison with other KGE algorithms or related methods.